# Bearing Syntactic Fruit with Stack-Augmented Neural Networks

## Abstract

Any finite set of training data is consistent with an infinite number of hypothetical algorithms that could have generated it. Studies have shown that when human children learn language, they consistently favor hypotheses based on hierarchical syntactic rules without ever encountering disambiguating examples. A recent line of work has inquired as to whether common neural network architectures share this bias, finding that they do so only under special conditions: when augmented with ground-truth parse tree structures, when pre-trained on massive corpora, or when trained long past convergence. In this paper, we demonstrate, for the first time, neural network architectures that are able to generalize in human-like fashion when trained only on surface forms and without any of the aforementioned requirements: stack-augmented neural networks. We test three base architectures (transformer, simple RNN, LSTM) augmented with two styles of stack: the superposition stack of Joulin & Mikolov (2015) and a nondeterministic generalization of it proposed by DuSell & Chiang (2023). We find that transformers with nondeterministic stacks generalize best out of these architectures on a classical question formation task. We also propose a modification to the stack RNN architecture that improves hierarchical generalization. These results suggest that stack-augmented neural networks may be more accurate models of human language acquisition than standard architectures, serving as useful objects of psycholinguistic study. Our code is publicly available.[1]

## 1 Introduction

When observing a finite set of input-output pairs for a task, a learner must anticipate unseen inputs by inferring a plausible algorithm that could have generated the outputs from the inputs. Multiple algorithms are always possible, and the learner must choose one of them. Consider, for instance, the task of **question formation**, i.e., converting a declarative sentence to its equivalent question form.

| | | |
|---|---|---|
| The salamanders don't amuse my newt. | $\rightarrow$ | Don't the salamanders amuse my newt? |
| My walrus does move. | $\rightarrow$ | Does my walrus move? |
| My orangutans do comfort the ravens. | $\rightarrow$ | Do my orangutans comfort the ravens? |

All of the examples above are compatible with the two algorithms in Figure 1 (at least). The **MOVE-MAIN** algorithm is linguistically correct, but neither algorithm is incorrect with respect to the examples above alone. The two would only differ from each other on an input like the following.

> The salamanders who do sleep don't amuse my newt.
> $\rightarrow$ Don't the salamanders who do sleep amuse my newt?  (**MOVE-MAIN**)
> $\rightarrow$ Do the salamanders who sleep don't amuse my newt?  (**MOVE-FIRST**)

Evaluating a learner on such an input would reveal its preference for **MOVE-MAIN**, **MOVE-FIRST**, or some other algorithm; we call this preference the learner's **inductive bias**.

The **MOVE-MAIN** algorithm requires an assumption that language conforms to hierarchical syntax, whereas **MOVE-FIRST** simply relies on linear position. Arguably, in the absence of disambiguating

---

[1]See the anonymous supplementary material.

**MOVE-MAIN**: Parse the sentence and front the main auxiliary verb.

**MOVE-FIRST**: Front the first auxiliary verb.

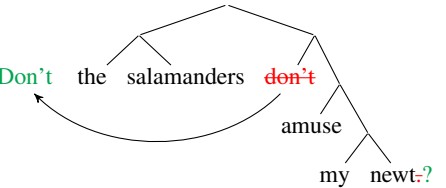

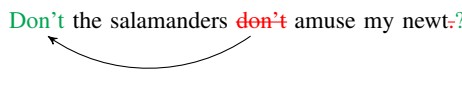

Figure 1: Two algorithms consistent with the training set for question formation.

examples like the above, **MOVE-FIRST** is simpler and so deserves preference. However, linguistic studies have shown that human children consistently form questions according to **MOVE-MAIN** (Crain & Nakayama, 1987), even though it is extremely unlikely that they have encountered disambiguating examples like the above (though some, like Pullum & Scholz (2002), have disputed this latter claim). Chomsky argued that this phenomenon reveals an inherent bias of human learners for hierarchical syntax—the so-called "argument from the poverty of the stimulus" (1965; 1980).

McCoy et al. (2020) investigated whether commonly used neural network architectures exhibit a similar bias. Their work and several follow-ups (Mulligan et al., 2021; Petty & Frank, 2021; Mueller et al., 2022; Mueller & Linzen, 2023; Yedetore et al., 2023; Murty et al., 2023; Mueller et al., 2024; Yedetore & Kim, 2024; Ginn, 2024; Ahuja et al., 2025) have answered this largely in the negative, except under special circumstances: namely, when a model is augmented with ground-truth parse trees (McCoy et al., 2020), pre-trained on implausibly large corpora (Mueller et al., 2022; Yedetore et al., 2023), or trained long past convergence on the validation data (Murty et al., 2023; Ahuja et al., 2025). McCoy et al. (2020) needed to go to rather extreme lengths to achieve hierarchical generalization; they used a Tree-LSTM encoder-decoder whose computation graph was structured according to the correct parse tree, not only on the input side, but also on the *output* side. On the other hand, they showed that merely including brackets in the data marking the correct parse tree structure did not cause standard architectures to generalize hierarchically. The ON-LSTM, a syntactically unsupervised architecture designed to handle bounded nesting depth, also failed to generalize. It has remained an open question whether there is a neural network architecture capable of generalizing in human-like fashion without requiring such strong syntactic supervision; as McCoy et al. (2020) put it, "Does syntax need to grow on trees?"

In this paper, we explore this question on a previously unstudied type of architecture: stack-augmented neural networks. Stack-augmented neural networks are sequential models that consist of a standard base architecture (e.g., an RNN or transformer) connected to a differentiable stack data structure (Joulin & Mikolov, 2015; DuSell & Chiang, 2022; 2023; 2024). Stacks are a cornerstone of context-free parsing algorithms and so are a natural choice for imbuing architectures with a bias for hierarchical syntax. We show that some of these stack-augmented architectures do, in fact, show a clear preference for hierarchical generalization on the question formation task without any of the special conditions listed above, often fronting not only the correct verb, but generating the entire output in accordance with **MOVE-MAIN** (up to 32% of the time). In this way, we show that one can "bear syntactic fruit" without explicit parse trees. Along the way, we also propose a modification to the stack-augmented RNN architecture that improves hierarchical generalization.

Our findings contribute to a broader debate about whether the acquisition of hierarchical generalization emanates from the learner (i.e., physiological biases in the human brain, as Elman et al. (1996) suggested) or from the stimulus. In a sense, a learning algorithm's bias always indicates a property of the *learner*; for any training set, one can adversarially construct a learning algorithm that ignores any hints one way or the other. Moreover, the notion of "simplicity" is always relative to the learner's parameterization. The question, then, is really whether a reasonably simple learning algorithm—not the kind of contrived example just mentioned, but perhaps a neural network architecture with minimal assumptions about the specific task—can learn a rule like **MOVE-MAIN** from ambiguous data while still attaining competitive performance on natural language benchmarks. We offer stack-augmented neural networks as a positive example that hierarchical generalization can emerge without explicit cues in the training data.

| my raven does change . DECL | → | my raven does change . |
| my raven does change . QUEST | → | does my raven change ? |
| my raven that doesn't sleep does change . QUEST | → | does my raven that doesn't sleep change ? |

(a) Question Formation

| our zebra changed . PAST | → | our zebra changed . |
| our zebra changed . PRESENT | → | our zebra changes . |
| our zebra below the vultures changed . PRESENT | → | our zebra below the vultures changes . |

(b) Tense Reinflection

Figure 2: Examples that illustrate the difference between the training (shaded white) and generalization (shaded gray) sets of each task.

## 2 TESTING THE POVERTY OF THE STIMULUS

The purpose of our experiments is to test whether a given neural network architecture, when trained on data that ambiguously conforms to rules based on hierarchical syntax and linear position, generalizes in accordance with the hierarchical rule. We use an established experimental framework to do so (Wilson, 2006; Frank & Mathis, 2007; McCoy et al., 2018), specifically using the **question formation** and **tense reinflection** tasks of McCoy et al. (2020). Both tasks involve transforming an input sentence to an output sentence. All input sentences were sampled from a simple probabilistic context-free grammar with a small vocabulary. Each task has a **training set** of 100k examples, a **validation set** of 1k examples, and an in-distribution **test set** of 10k examples where all examples are consistent with both the hierarchical and linear rule. Each task also has an out-of-distribution **generalization set** of 10k examples that is consistent only with the hierarchical rule; evaluating on this dataset reveals the model's inductive bias.

As described in the introduction, question formation entails transforming a declarative sentence into the equivalent yes-no question (Figure 2a). The input sentence always ends in a special token that indicates one of two types of transformation to apply: DECL, which means to copy the declarative sentence to the output unchanged; and QUEST, which means to convert it to a question. In examples with DECL, the subject noun phrase may or may not be followed by a relative clause containing a verb, so the main verb is not necessarily the first verb in the sentence. On the other hand, in examples with QUEST, the subject noun phrase is never followed by a relative clause containing a verb, so the main verb always happens to be the first verb in the sentence. Therefore, the training data is consistent with both **MOVE-MAIN** and **MOVE-FIRST**. The generalization set consists entirely of examples that contain QUEST where the subject noun phrase is followed by a relative clause with a verb, in which case only **MOVE-MAIN** correctly predicts the output.

Tense reinflection entails transforming all verbs in a sentence from past tense to present tense (Figure 2b). The input sentence ends in a special token indicating one of two transformations to apply: PAST, which means to copy the past-tense sentence to the output unchanged; and PRESENT, which means to convert all verbs to present tense. In general, the form of each verb must agree in number with its subject. In past tense, verbs do not inflect for person and number; but in present tense, they do. Inflecting the verb correctly requires identifying its grammatical subject and its number. Each sentence contains a main verb, and the subject and object may be followed by prepositional phrases containing another noun. In the training data, for examples with PAST, the subject nouns and the nouns in prepositional phrases that modify them may differ in number. On the other hand, for examples with PRESENT, their numbers are always the same. Therefore, the training data is consistent with the two algorithms in Figure 3 (at least). The generalization set consists entirely of examples with PRESENT where a prepositional phrase modifies the subject noun phrase and contains a noun that differs in number, in which case only **AGREE-MAIN** predicts the output correctly.

## 3 DIFFERENTIABLE STACKS

We now review stack-augmented neural network architectures. Stacks are a central component of many context-free grammar parsing algorithms, as their last-in-first-out protocol makes them suited for tracking unclosed constituents (in top-down parsing) or completed constituents (in bottom-up parsing) in the proper order. Stack-augmented neural network architectures consist of a standard

**AGREE-MAIN**: Parse the sentence and inflect each verb to agree with its subject noun.

**AGREE-RECENT**: Inflect each verb to agree with its most recent noun.

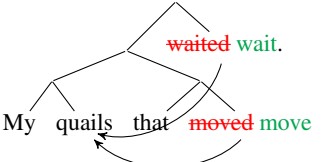

My quails that ~~moved~~ move ~~waited~~ wait.

Figure 3: Two algorithms consistent with the training set for tense reinflection.

base architecture (i.e., a simple RNN, LSTM, or transformer) connected to a **differentiable stack**. We first explain differentiable stacks and later explain how they interface with the base architecture.

A differentiable stack is a continuous function that simulates the behavior of a discrete stack; there are multiple kinds, which we will discuss below. Conceptually, a differentiable stack converts the familiar operations of pushing, popping, and reading the top element to continuous relaxations. Instead of receiving a single push or pop command at a time, it receives a set of weighted **stack actions**, which it simulates in proportion to their weights. The differentiable stack then produces a **stack reading** that represents an interpolation of the topmost stack element after applying those action weights. A neural network can compute the stack actions dynamically in one part and use the stack reading as input in another part. Because the stack reading is differentiable with respect to the stack actions, the whole network can still be trained end-to-end with ordinary backpropagation and gradient descent—and without needing supervision over the stack actions. In this way, the differentiable stack provides the model with a latent representation of syntactic structure.

For both types of differentiable stack that we use in this paper, the differentiable stack simulates a discrete stack whose elements are vectors in $(0, 1)^m$, where $m$ is a hyperparameter. The differentiable stack starts in an initial, empty state $\mathcal{S}_0$. We can iteratively apply actions to the differentiable stack and extract a stack reading at each step. At iteration $t > 0$, let $\boldsymbol{a}_t \in \mathbb{R}_+^a$ be a vector of stack action weights, let $\boldsymbol{a}_t' \in \mathbb{R}^a$ be a vector of unnormalized logits used to compute $\boldsymbol{a}_t$, let $\boldsymbol{v}_t \in (0, 1)^m$ be the vector pushed to the stack by the push action, and let $\boldsymbol{r}_t \in \mathbb{R}_+^r$ be the resulting stack reading. We can abstract the differentiable stack into three functions: ACTIONS, which converts $\boldsymbol{a}_t'$ to $\boldsymbol{a}_t$; STACK, which incrementally updates the stack; and READING, which produces the stack reading.

$$\boldsymbol{a}_t \stackrel{\text{def}}{=} \text{ACTIONS}(\boldsymbol{a}_t') \qquad \mathcal{S}_t \stackrel{\text{def}}{=} \text{STACK}(\mathcal{S}_{t-1}, \boldsymbol{a}_t, \boldsymbol{v}_t) \qquad \boldsymbol{r}_t \stackrel{\text{def}}{=} \text{READING}(\mathcal{S}_t) \qquad (1)$$

### 3.1 SUPERPOSITION STACK

The **superposition stack** of Joulin & Mikolov (2015) assumes that the action vector is a probability distribution over three actions: push, no-op, and pop. We have $\boldsymbol{a}_t \stackrel{\text{def}}{=} (a_t^{\text{PUSH}}, a_t^{\text{NOOP}}, a_t^{\text{POP}}) \stackrel{\text{def}}{=} \text{softmax}(\boldsymbol{a}_t')$. The stack $\mathcal{S}_t$ is a matrix $\boldsymbol{V}_t \in \mathbb{R}^{(t+1) \times m}$, where the vector $(\boldsymbol{V}_t)_i$ represents the $i^{\text{th}}$ element from the top of the stack. The initial stack, $\boldsymbol{V}_0$, contains a single $\boldsymbol{0}$ vector. The function STACK works by computing three new, separate stacks: one with all elements shifted down and $\boldsymbol{v}_t$ inserted at the top (push); one kept the same (no-op); and one with all elements shifted up and the topmost deleted (pop). The new stack is formed by interpolating these stacks elementwise according to the push, no-op, and pop probabilities, making a superposition of the three. The function READING simply returns the top vector in $\boldsymbol{V}_t$. DuSell & Chiang (2024) pointed out that, mathematically, the stack reading at step $t$ is simply a summation over all possible sequences of stack actions ending at $t$, so it resembles a kind of structured attention mechanism over $\boldsymbol{v}_1, \ldots, \boldsymbol{v}_t$ based on syntactic structures. For details, we refer the reader to §A.1 and DuSell & Chiang (2024).

### 3.2 NONDETERMINISTIC STACK

The **nondeterministic stack** of DuSell & Chiang (2020; 2022; 2023) simulates a nondeterministic pushdown automaton (PDA) and is a generalization of the superposition stack. More precisely, we use what DuSell & Chiang (2024) call the **differentiable vector PDA (dVPDA)**. The dVPDA is a continuous function that simulates a modified PDA called a **vector PDA (VPDA)**. Like a traditional

PDA, a VPDA consists of a finite set of states $Q$, a finite set of transitions, and an infinite stack data structure. Elements of the stack are members of $\Gamma \times (0, 1)^m$, where $\Gamma$ is a finite alphabet of stack symbols. The two parts of these stack elements serve different roles: the discrete symbol from $\Gamma$ interacts with the PDA's state machine, whereas the vector from $(0, 1)^m$ serves as an efficient way to encode information tacked onto each stack element. In order to make the simulation tractable, the VPDA's state machine cannot observe these vectors; it can only push an externally provided vector $\boldsymbol{v}_t$ onto the stack and expose previously pushed vectors at the top of the stack.

Let $\boldsymbol{w} = w_1 \cdots w_n$ be a string of input symbols. The VPDA starts in an initial state $q_0 \in Q$ with a stack that contains only $(\bot, \boldsymbol{v}_0)$, where $\bot \in \Gamma$ is a designated bottom symbol, and $\boldsymbol{v}_0$ is a learned initial bottom vector. The VPDA has three types of transition, where $q, r \in Q$ and $x, y \in \Gamma$, with the following semantics.

$$
\begin{array}{lll}
q, x \xrightarrow{w_t} r, xy & \text{If } (x, \boldsymbol{u}) \text{ is on top, } \textbf{push } (y, \boldsymbol{v}_t). \\
q, x \xrightarrow{w_t} r, y & \text{If } (x, \boldsymbol{u}) \text{ is on top, } \textbf{replace } \text{it with } (y, \boldsymbol{u}). \\
q, x \xrightarrow{w_t} r, \varepsilon & \text{If } (x, \boldsymbol{u}) \text{ is on top, } \textbf{pop } \text{it.}
\end{array}
$$

A *deterministic* VPDA would only admit at most one outgoing transition for each state $q$ and top stack symbol $x$. In contrast, the dVPDA simulates a *nondeterministic* VPDA that allows multiple, in which case it simulates all possible sequences of transitions. In other words, the dVPDA does not have to commit to a single parse of its input while incrementally processing it. Rather, it can encode a *distribution* over all possible syntactic structures. The superposition stack is like a special case of dVPDA where $Q = \{q_0\}$, $\Gamma = \{\bot\}$, and $\boldsymbol{v}_0 = \boldsymbol{0}$. For details, we refer the reader to §A.2 and DuSell & Chiang (2023).

## 4 STACK-AUGMENTED NEURAL NETWORKS

Here, we describe how to incorporate differentiable stacks into three base architectures: the simple RNN (Elman, 1990), LSTM (Hochreiter & Schmidhuber, 1997), and transformer (Vaswani et al., 2017). For simplicity, we will use one free hyperparameter $d$ to control the size of the model and express other hyperparameters in terms of $d$. For the RNN and LSTM, $d$ is the number of hidden units; for the transformer, $d$ is the model width $d_{\text{model}}$. Let $\Sigma$ be a finite vocabulary of tokens, and let $\boldsymbol{w} = w_1 \cdots w_n \in \Sigma^n$ be the input string. Assume the model is a language model that computes logit vectors $\boldsymbol{y}_0, \ldots, \boldsymbol{y}_n \in \mathbb{R}^{|\Sigma|+1}$, where for each $0 \le t \le n$, $b \in \Sigma \cup \{\text{EOS}\}$, the next-token probability $p_M(b \mid w_1 \cdots w_t) \stackrel{\text{def}}{=} \text{softmax}(\boldsymbol{y}_t)_b$. In the following, $\sigma(\cdot)$ is the logistic function, and DROPOUT$(\cdot)$ indicates the application of dropout (Srivastava et al., 2014).

### 4.1 STACK RNN AND LSTM

RNNs and LSTMs connect to differentiable stacks in the same way. We make two minor innovations compared to prior work: we use a base RNN or LSTM with multiple layers, and we add dropout. First, we describe how the base architectures work. Let $L$ be the number of layers. At each step $t$, the model computes a series of hidden states $\boldsymbol{h}_t^{(1)}, \ldots, \boldsymbol{h}_t^{(L)}$. (In the case of the LSTM, it also computes a series of memory cell values.) Let us encapsulate the entire state of the model at step $t$ in a single object $\mathcal{H}_t$, and let RECURRENCE be the function that updates the state. Each architecture has an initial state $\mathcal{H}_0$. Let $\boldsymbol{E} \in \mathbb{R}^{(|\Sigma|+1) \times d}$ be a learned embedding matrix, and let $\boldsymbol{x}_t \stackrel{\text{def}}{=} \boldsymbol{E}_{w_t}$ be the input embedding of $w_t$. We apply dropout like Zaremba et al. (2015), and we tie the input and output embeddings (Press & Wolf, 2017). For $0 < t \le n$, $\boldsymbol{x}_t' \stackrel{\text{def}}{=} \text{DROPOUT}(\boldsymbol{x}_t)$, $\mathcal{H}_t \stackrel{\text{def}}{=} \text{RECURRENCE}(\mathcal{H}_{t-1}, \boldsymbol{x}_t')$, $\boldsymbol{y}_t' \stackrel{\text{def}}{=} \text{DROPOUT}(\boldsymbol{h}_t^{(L)})$, and $\boldsymbol{y}_t \stackrel{\text{def}}{=} \boldsymbol{E}\boldsymbol{y}_t'$. See §§ B.1 and B.2 for details.

To connect a base RNN or LSTM, called the **controller**, to a differentiable stack, we use the last layer's hidden state to compute the action logits as $\boldsymbol{a}_t' \stackrel{\text{def}}{=} \boldsymbol{W}_{\text{a}} \boldsymbol{h}_t^{(L)} + \boldsymbol{b}_{\text{a}}$ and the pushed vector as $\boldsymbol{v}_t \stackrel{\text{def}}{=} \sigma(\boldsymbol{W}_{\text{v}} \boldsymbol{h}_t^{(L)} + \boldsymbol{b}_{\text{v}})$, where $\boldsymbol{W}_{\text{a}} \in \mathbb{R}^{a \times d}$, $\boldsymbol{b}_{\text{a}} \in \mathbb{R}^a$, $\boldsymbol{W}_{\text{v}} \in \mathbb{R}^{m \times d}$, and $\boldsymbol{b}_{\text{v}} \in \mathbb{R}^m$ are learned parameters. We then use the stack reading as an extra input to the next update, giving $\mathcal{H}_t \stackrel{\text{def}}{=} \text{RECURRENCE}(\mathcal{H}_{t-1}, \begin{bmatrix} \boldsymbol{x}_t' \\ \boldsymbol{r}_t \end{bmatrix})$. See Figure 4 for an illustration.

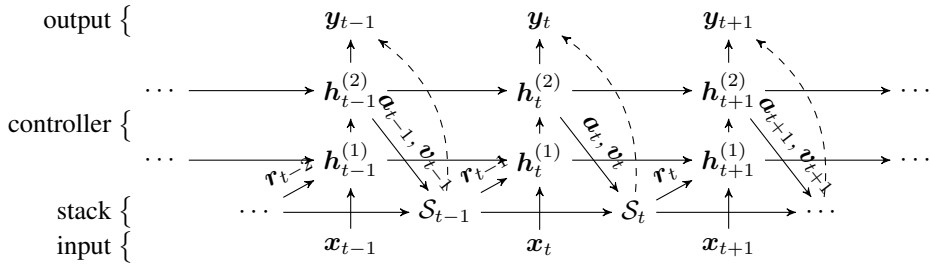

Figure 4: Conceptual diagram of the controller-stack interface for RNNs and LSTMs, unrolled across a portion of time and with $L = 2$ layers. The dashed lines indicate our proposed short-circuit connection from the stack reading to the output.

Note that in this setup, when the hidden state issues actions at step $t$ that update the stack, it cannot access the updated stack reading until step $t + 1$. This essentially causes an off-by-one lag between the model's predictions and stack state. Indeed, the proof of DuSell & Chiang's (2023) Prop. 1, which shows that nondeterministic stack RNNs can recognize all context-free languages, relies on a final EOS symbol to work around this. Although it may be possible for a model to learn a workaround, doing so likely overcomplicates simple parsing tasks and hinders hierarchical generalization. We propose the following fix. At step $t$, we simply short-circuit $\boldsymbol{r}_t$ to the output $\boldsymbol{y}'_t$ by concatenating it to the final layer's hidden state. More precisely, $\boldsymbol{y}'_t \stackrel{\text{def}}{=} \text{DROPOUT}(\begin{bmatrix} \boldsymbol{h}_t^{(L)} \\ \boldsymbol{r}_t \end{bmatrix})$. We also adjust the size of the embedding vectors in $\boldsymbol{E}$ to match the new size of $\boldsymbol{y}'_t$, so that $\boldsymbol{E} \in \mathbb{R}^{(|\Sigma|+1) \times (d+r)}$, where $r$ is the size of $\boldsymbol{r}_t$.

### 4.2 STACK ATTENTION TRANSFORMER

A transformer encoder consists of multiple layers, each of which consists of an attention sublayer followed by a feedforward sublayer. Since the differentiable stacks described earlier resemble structured attention mechanisms over syntactic structures, DuSell & Chiang (2024) proposed incorporating them into the transformer by swapping them in place of the scaled dot-product attention operator in some of the attention sublayers. Let $\bar{\boldsymbol{x}}_t$ be the input to a sublayer. In a transformer with pre-norm, we pass the input through layer norm (Ba et al., 2016) to get $\bar{\boldsymbol{x}}_t^{\text{LN}} \stackrel{\text{def}}{=} \text{LAYERNORM}(\bar{\boldsymbol{x}}_t)$. Let SUBLAYER be the **sublayer function**, which encapsulates the core behavior of the sublayer, such as attention or a feedforward layer. Then $\bar{\boldsymbol{y}}_t^{\text{SL}} \stackrel{\text{def}}{=} \text{SUBLAYER}(t, (\bar{\boldsymbol{x}}_1^{\text{LN}}, \ldots, \bar{\boldsymbol{x}}_n^{\text{LN}}))$, and the output is computed by applying dropout to the sublayer output and adding a residual connection, giving $\bar{\boldsymbol{y}}_t \stackrel{\text{def}}{=} \bar{\boldsymbol{x}}_t + \text{DROPOUT}(\bar{\boldsymbol{y}}_t^{\text{SL}})$.

A stack attention layer implements SUBLAYER as follows. We compute the stack action logits as $\boldsymbol{a}'_t \stackrel{\text{def}}{=} \boldsymbol{W}_{\text{a}} \bar{\boldsymbol{x}}_t^{\text{LN}}$ and the pushed vector as $\boldsymbol{v}_t \stackrel{\text{def}}{=} \sigma(\boldsymbol{W}_{\text{v}} \bar{\boldsymbol{x}}_t^{\text{LN}})$, where $\boldsymbol{W}_{\text{a}} \in \mathbb{R}^{a \times d}$ and $\boldsymbol{W}_{\text{v}} \in \mathbb{R}^{m \times d}$ are learned parameters. The sublayer starts with an empty stack and applies the stack operations on it for $t = 1, \ldots, n$. The stack returns a stack reading $\boldsymbol{r}_t$ at each step, and we compute the sublayer output as $\bar{\boldsymbol{y}}_t \stackrel{\text{def}}{=} \boldsymbol{W}_{\text{y}} \boldsymbol{r}_t$, where $\boldsymbol{W}_{\text{y}} \in \mathbb{R}^{d \times r}$ is a learned parameter. We illustrate this in Figure 5.

## 5 EXPERIMENTS

We train stack-augmented and vanilla neural networks to transform an input string $\boldsymbol{x}$ to an output string $\boldsymbol{y}$ for both the question formation and tense reinflection tasks. Following Murty et al. (2023) and Ahuja et al. (2025), we train all models as autoregressive language models on the concatenation of the input and output, $\boldsymbol{w} = \boldsymbol{x}\boldsymbol{y}$. Prompting a model $M$ with a prefix $\boldsymbol{x}$ allows us to sample outputs from or measure the probabilities of outputs in the conditional distribution $p_M(\cdot \mid \boldsymbol{x})$.

We automatically adjust $d$ for each model so that its total parameter count is as close as possible to 200k, ensuring that all models are of comparable size. We use SymPy (Meurer et al., 2017) to work out the algebra automatically. All RNNs and LSTMs have 3 layers, and all transformers have

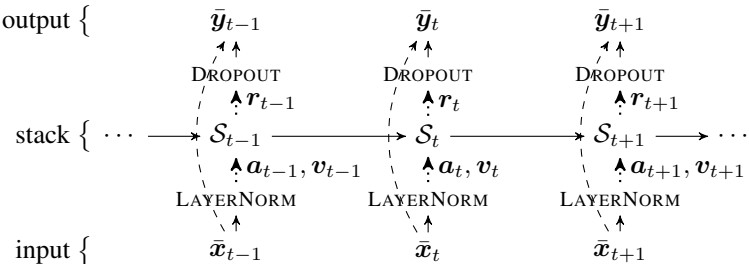

Figure 5: Conceptual diagram of a stack attention sublayer, unrolled across a portion of time. Dotted arrows indicate linear transformations, and dashed arrows indicate residual connections.

5 layers. We use transformers with one or two stack attention layers. For transformers with one stack attention layer, we swap it into the attention mechanism of the third layer. For transformers with two stack attention layers, we swap them into the second and fourth layers. All superposition stacks use a stack vector size of $m = 50$. All dVPDAs use $|Q| = 3$, $|\Gamma| = 3$, and $m = 5$.

We have included transformers with two stack attention layers for the following reason. Even with a stack, an autoregressive language model is still somewhat at a disadvantage on these tasks, in that the model not only needs to parse the input, but also to continue generating a transformed version of it. This is akin to modeling cross-serial dependencies, for which a stack is inadequate; if the language of inputs admitted arbitrary levels of recursion, then the language of concatenated inputs and outputs would not even be context-free. DuSell & Chiang (2023) showed that nondeterministic stack LSTMs can actually learn cross-serial dependencies, but stack transformers cannot perform the same trick with a single layer. Thus, it might be helpful for the transformer to have a stack attention layer that parses the input, an intermediate standard attention layer that copies it to the output, and another stack attention layer that models the syntax of the output.

We optimize each model using a standard cross-entropy objective. For each architecture, we do an initial hyperparameter search by training 10 models with randomly sampled hyperparameters for a maximum of 5 epochs each. We select the hyperparameters that result in the lowest cross-entropy on the validation set and then train 5 models to convergence. We report the mean and standard deviation of the performance of these models. See App. C for details.

We also make some improvements upon previously used metrics. Instead of evaluating the accuracy of greedily decoded outputs, we measure *probabilities* of outputs, giving us a sense of the expected accuracy with respect to $p_M(\cdot \mid \boldsymbol{x})$. First, we report the average **conditional probability (CP)** $p_M(\boldsymbol{y} \mid \boldsymbol{x})$, which we can compute exactly; this is equivalent to the expectation of the full-sentence accuracy when using ancestral sampling. Because the transformation is deterministic conditioned on the input, a probability of 1 is achievable. Let $D$ be a dataset of input-output pairs.

$$\mathrm{CP}(M, D) \stackrel{\text{def}}{=} \frac{1}{|D|} \sum_{(\boldsymbol{x}, \boldsymbol{y}) \in D} p_M(\boldsymbol{y} \mid \boldsymbol{x}) = \frac{1}{|D|} \sum_{(\boldsymbol{x}, \boldsymbol{y}) \in D} \mathbb{E}_{\boldsymbol{y}' \sim p_M(\cdot | \boldsymbol{x})} \mathbb{1}[\boldsymbol{y}' = \boldsymbol{y}] \tag{2}$$

In order to compare models that have near-zero CP, we also report the **conditional cross-entropy (CCE)** of the output, micro-averaged by the number of words in $\boldsymbol{y}$ plus EOS. Again, because the transformation is deterministic, a score of 0 is achievable.

$$\mathrm{CCE}(M, D) \stackrel{\text{def}}{=} \frac{\sum_{(\boldsymbol{x}, \boldsymbol{y}) \in D} -\log p_M(\boldsymbol{y} \mid \boldsymbol{x})}{\sum_{(\boldsymbol{x}, \boldsymbol{y}) \in D} (|\boldsymbol{y}| + 1)} \tag{3}$$

We also use **fine-grained accuracy (FA)** metrics, which are more forgiving than exact match. These test whether the model identified the correct main verb without penalizing it for other mistakes. For question formation, this tests whether the output begins with the correct verb. For tense reinflection, this tests whether the output is of the correct length, has words that are all of the correct parts of speech in the correct positions (even if the exact word choices do not match), and that the surface form of the reinflected main verb is correct. We compute both in expectation over $p_M(\cdot \mid \boldsymbol{x})$. For question formation, we can compute the probability of the first word exactly by looking at

the conditional distribution over words after reading $\boldsymbol{x}$. For tense reinflection, for each input, we randomly sample 10 outputs and use those to estimate the expected fine-grained accuracy.

We also compute the **log ratio (LR)** of the conditional probability that the model assigns to the outputs predicted by the hierarchical rule to that of the linear rule. We express this as a log for readability. Positive values indicate a preference for the hierarchical rule; and negative, for the linear. Let $D_{\text{gen}}$ be the generalization set, and let $\boldsymbol{y}_{\text{lin}}$ be the counterpart to output $\boldsymbol{y}$ predicted by the linear rule.

$$\text{LR}(M) \stackrel{\text{def}}{=} \frac{1}{|D_{\text{gen}}|} \sum_{(\boldsymbol{x}, \boldsymbol{y}) \in D_{\text{gen}}} \log \frac{p_M(\boldsymbol{y} \mid \boldsymbol{x})}{p_M(\boldsymbol{y}_{\text{lin}} \mid \boldsymbol{x})} \quad (4)$$

## 6 RESULTS

Our results are in Tables 1 and 2. For both tasks, only transformers learn the in-distribution test set. The RNNs and LSTMs likely fail to learn the test set because their hidden states, which are of constant size, impose an information bottleneck between the input and output, hindering the models from copying parts of the input to the output faithfully. The transformers do not have this bottleneck thanks to their self-attention mechanism.[2] For question formation, the transformer with nondeterministic stack attention (Tf+Nd) attains the best CP on the generalization set, generating the correct full output about 32% of the time, an improvement from the transformer's (Tf) meager 0.5%. Adding a stack to Tf improves CP in all cases. Furthermore, Tf+Nd and Tf+Nd+Nd attain the highest FA (up to 86%) and a very high ratio in favor of the hierarchical generalization. This surpasses the 78% FA of McCoy et al.'s (2020) GRU with location-based attention and the 5% of their ON-LSTM (Shen et al., 2019), another stack-based model. It also surpasses the approximately 76% FA of Ahuja et al.'s (2025) overtrained transformer language model. The RNNs consistently favor the linear generalization even with stacks, but the LSTMs favor the hierarchical generalization. Our short-circuit trick for the stack reading (+R) in stack RNNs and LSTMs consistently tips the ratio toward hierarchical generalization. The best non-transformer model is LSTM+Nd+R.

The results for tense reinflection, however, are quite different: no models, not even the stack-augmented ones, consistently generalize hierarchically or prefer the hierarchical output. Prior work has similarly shown weaker hierarchical generalization on this task (McCoy et al., 2020; Mulligan et al., 2021); Mulligan et al. (2021) pointed out that tense reinflection is not a linguistically natural task, as the past is typically not seen as a base form and the present as a transformation of it. McCoy et al.'s (2020) best syntactically unsupervised model was the ON-LSTM, with an FA of 5%. In our results, Tf already improves on this with 9%, and Tf+Nd+Nd increases it to 10%. Some of the stack RNNs and LSTMs attain much higher FA, but they do not even learn the test distribution. Tf already generates the whole output correctly 8% of the time; only Tf+Nd+Nd has a higher generalization CP, with 9%. We see that +R helps the ratio of the RNN, but not the LSTM. RNN+Sup+R and LSTM+Nd have notably high FA and log ratios but still prefer the linear generalization.

## 7 CONCLUSION

Our results show that on question formation, transformers with nondeterministic stack attention show a much stronger hierarchical bias than standard architectures, putting them more in line with human biases. DuSell & Chiang (2024) showed that the same architecture outperforms the vanilla transformer on a Penn Treebank language modeling task despite having fewer parameters, so the nondeterministic stack attention transformer not only generalizes more naturally, but fits in-distribution natural language better, suggesting that it is a promising object of future psycholinguistic study. We showed that short-circuiting the stack reading to the output improves generalization performance for RNNs and LSTMs, particularly for LSTMs, although not to the level of the stack transformer. Our findings, however, do not generalize to the tense reinflection task, and it remains to be seen if a single architecture can generalize hierarchically on both tasks.

---

[2]Unlike our RNN and LSTM language models, which do not have self-attention, the RNN, GRU, and LSTM models of McCoy et al. (2020) were encoder-decoder models with attention, hence they were sometimes able to learn the test set.

Table 1: Results on question formation. Each row is the mean of 5 runs with standard deviations in small text and the best mean value of each column in **bold**. "Tf" = transformer; "RNN" = simple RNN; "LSTM" = LSTM; "+Sup" = with superposition stack; "+Nd" = with nondeterministic stack; "+R" = with the stack reading short-circuited to the output. "CP" = conditional probability; "CCE" = conditional cross-entropy; "FA" = fine-grained accuracy. *Results from McCoy et al. (2020).

| | Test | Generalization | | | |
| --- | --- | --- | --- | --- | --- |
| Architecture | CP ↑ | CP ↑ | CCE ↓ | FA ↑ | Log Ratio ↑ |
| Tf | $\mathbf{0.999}_{\pm.00}$ | $0.005_{\pm.01}$ | $1.493_{\pm.11}$ | $0.645_{\pm.25}$ | $1.724_{\pm2.12}$ |
| Tf+Sup | $0.999_{\pm.00}$ | $0.039_{\pm.04}$ | $1.702_{\pm.44}$ | $0.325_{\pm.21}$ | $1.062_{\pm11.80}$ |
| Tf+Sup+Sup | $0.998_{\pm.00}$ | $0.161_{\pm.18}$ | $1.498_{\pm.33}$ | $0.697_{\pm.08}$ | $1.844_{\pm5.29}$ |
| Tf+Nd | $0.999_{\pm.00}$ | $\mathbf{0.318}_{\pm.19}$ | $\mathbf{1.285}_{\pm.42}$ | $0.732_{\pm.15}$ | $\mathbf{20.506}_{\pm6.93}$ |
| Tf+Nd+Nd | $0.995_{\pm.00}$ | $0.191_{\pm.22}$ | $1.336_{\pm.64}$ | $\mathbf{0.862}_{\pm.17}$ | $17.717_{\pm11.16}$ |
| RNN | $0.465_{\pm.31}$ | $0.000_{\pm.00}$ | $3.216_{\pm.09}$ | $0.659_{\pm.20}$ | $-5.106_{\pm1.33}$ |
| RNN+Sup | $0.022_{\pm.02}$ | $0.000_{\pm.00}$ | $3.469_{\pm.22}$ | $0.417_{\pm.16}$ | $-5.066_{\pm2.92}$ |
| RNN+Sup+R | $0.011_{\pm.01}$ | $0.000_{\pm.00}$ | $3.991_{\pm.13}$ | $0.482_{\pm.02}$ | $-4.628_{\pm2.66}$ |
| RNN+Nd | $0.027_{\pm.02}$ | $0.000_{\pm.00}$ | $3.422_{\pm.07}$ | $0.525_{\pm.07}$ | $-6.308_{\pm1.36}$ |
| RNN+Nd+R | $0.019_{\pm.01}$ | $0.000_{\pm.00}$ | $4.039_{\pm.32}$ | $0.511_{\pm.03}$ | $-5.307_{\pm1.98}$ |
| LSTM | $0.003_{\pm.00}$ | $0.000_{\pm.00}$ | $1.501_{\pm.32}$ | $0.497_{\pm.00}$ | $6.873_{\pm5.57}$ |
| LSTM+Sup | $0.002_{\pm.00}$ | $0.000_{\pm.00}$ | $2.027_{\pm.44}$ | $0.498_{\pm.00}$ | $5.929_{\pm4.19}$ |
| LSTM+Sup+R | $0.005_{\pm.00}$ | $0.001_{\pm.00}$ | $1.384_{\pm.17}$ | $0.505_{\pm.02}$ | $10.281_{\pm6.18}$ |
| LSTM+Nd | $0.002_{\pm.00}$ | $0.000_{\pm.00}$ | $2.678_{\pm.90}$ | $0.499_{\pm.00}$ | $0.428_{\pm5.42}$ |
| LSTM+Nd+R | $0.118_{\pm.17}$ | $0.009_{\pm.02}$ | $1.960_{\pm.52}$ | $0.505_{\pm.07}$ | $13.204_{\pm11.61}$ |
| Tree-GRU* | 0.96 | – | – | 0.99 | – |
| ON-LSTM* | 0.93 | – | – | 0.05 | – |

Table 2: Results on tense reinflection. All labels have the same meanings as in Table 1.

| | Test | Generalization | | | |
| --- | --- | --- | --- | --- | --- |
| Architecture | CP ↑ | CP ↑ | CCE ↓ | FA ↑ | Log Ratio ↑ |
| Tf | $\mathbf{0.999}_{\pm.00}$ | $0.079_{\pm.06}$ | $0.397_{\pm.08}$ | $0.086_{\pm.07}$ | $-4.399_{\pm1.04}$ |
| Tf+Sup | $0.993_{\pm.01}$ | $0.036_{\pm.03}$ | $0.432_{\pm.09}$ | $0.048_{\pm.03}$ | $-4.876_{\pm1.14}$ |
| Tf+Sup+Sup | $0.984_{\pm.01}$ | $0.059_{\pm.04}$ | $\mathbf{0.364}_{\pm.08}$ | $0.061_{\pm.04}$ | $-4.063_{\pm.98}$ |
| Tf+Nd | $0.998_{\pm.00}$ | $0.056_{\pm.06}$ | $0.552_{\pm.17}$ | $0.062_{\pm.07}$ | $-6.196_{\pm2.08}$ |
| Tf+Nd+Nd | $0.997_{\pm.00}$ | $\mathbf{0.086}_{\pm.09}$ | $0.601_{\pm.22}$ | $0.101_{\pm.11}$ | $-6.667_{\pm2.71}$ |
| RNN | $0.051_{\pm.01}$ | $0.000_{\pm.00}$ | $1.069_{\pm.05}$ | $0.012_{\pm.01}$ | $-5.864_{\pm.45}$ |
| RNN+Sup | $0.024_{\pm.01}$ | $0.000_{\pm.00}$ | $1.207_{\pm.11}$ | $0.015_{\pm.01}$ | $-6.218_{\pm.90}$ |
| RNN+Sup+R | $0.017_{\pm.01}$ | $0.000_{\pm.00}$ | $1.256_{\pm.07}$ | $\mathbf{0.331}_{\pm.40}$ | $\mathbf{-2.973}_{\pm3.33}$ |
| RNN+Nd | $0.059_{\pm.03}$ | $0.000_{\pm.00}$ | $0.988_{\pm.16}$ | $0.028_{\pm.01}$ | $-6.387_{\pm.55}$ |
| RNN+Nd+R | $0.024_{\pm.02}$ | $0.000_{\pm.00}$ | $1.258_{\pm.15}$ | $0.037_{\pm.04}$ | $-5.577_{\pm1.80}$ |
| LSTM | $0.029_{\pm.03}$ | $0.000_{\pm.00}$ | $1.295_{\pm.25}$ | $0.067_{\pm.08}$ | $-7.877_{\pm1.59}$ |
| LSTM+Sup | $0.016_{\pm.00}$ | $0.000_{\pm.00}$ | $1.367_{\pm.11}$ | $0.041_{\pm.06}$ | $-7.300_{\pm1.04}$ |
| LSTM+Sup+R | $0.018_{\pm.01}$ | $0.000_{\pm.00}$ | $1.358_{\pm.24}$ | $0.050_{\pm.07}$ | $-7.718_{\pm1.23}$ |
| LSTM+Nd | $0.012_{\pm.01}$ | $0.000_{\pm.00}$ | $1.280_{\pm.22}$ | $0.243_{\pm.21}$ | $-3.762_{\pm3.22}$ |
| LSTM+Nd+R | $0.017_{\pm.01}$ | $0.000_{\pm.00}$ | $1.294_{\pm.31}$ | $0.086_{\pm.06}$ | $-6.601_{\pm1.17}$ |
| Tree-GRU* | 0.96 | – | – | 0.94 | – |
| ON-LSTM* | 0.95 | – | – | 0.05 | – |

REPRODUCIBILITY STATEMENT

To foster reproducibility, we have publicly released all of the code used to download the datasets we used and produce our experimental results. In order to simplify the replication of our software environment, during development and experimentation, we ran our code in containers, whose image definition we have included in our code. We have included the shell commands we used to produce each experiment and table. We have thoroughly documented our experimental methodology and settings in §5 and App. C.

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

## A  DETAILS OF DIFFERENTIABLE STACKS

### A.1  SUPERPOSITION STACK

Let $a_1, \ldots, a_t$ be a sequence of stack action vectors, and let $v_1, \ldots, v_t$ be the corresponding pushed vectors. Let a **run** $\pi = \tau_i, \ldots, \tau_j$, where $\tau_i, \ldots, \tau_j \in \{\text{PUSH}, \text{NOOP}, \text{POP}\}$, be any sequence of

actions that, if executed on an empty stack, would never attempt to pop the stack when it was empty. Let $\psi(\boldsymbol{\pi})$ be the weight of $\boldsymbol{\pi}$, which is the product $a_i^{\tau_i} \cdots a_j^{\tau_j}$. Let $\boldsymbol{v}(\boldsymbol{\pi})$ be the top stack element that would result from starting with a stack containing just $\mathbf{0}$ and executing the actions in $\boldsymbol{\pi}$ on it, using $\boldsymbol{v}_t$ as the pushed vector whenever $\tau_t = \text{PUSH}$. Let $\Pi[\text{PUSH} \rightsquigarrow t]$ be the set of all runs that start with a PUSH and end at step $t$. Then the stack reading can be expressed as

$$\boldsymbol{r}_t = \frac{\sum_{\boldsymbol{\pi} \in \Pi[\text{PUSH} \rightsquigarrow t]} \psi(\boldsymbol{\pi}) \, \boldsymbol{v}(\boldsymbol{\pi})}{\sum_{\boldsymbol{\pi} \in \Pi[\text{PUSH} \rightsquigarrow t]} \psi(\boldsymbol{\pi})}. \tag{5}$$

The procedure described in §3.1 is an efficient dynamic programming algorithm for computing Eq. (5). DuSell & Chiang (2024) pointed out that this resembles a kind of structured attention mechanism over $\boldsymbol{v}_1, \ldots, \boldsymbol{v}_t$ based on sequences of stack actions.

## A.2 Nondeterministic stack

In the dVPDA, each transition $\tau$ has a non-negative weight $\psi(\tau)$; accordingly, the action vector $\boldsymbol{a}_t = \exp(\boldsymbol{a}_t')$ contains the weights of all transitions corresponding to $w_t$, of which there are a fixed number. Let a **run** $\boldsymbol{\pi} = \tau_1, \ldots, \tau_j$ be any valid sequence of transitions that starts in the initial configuration and never fully empties the stack (replacing the bottom element does not count). Let $\psi(\boldsymbol{\pi})$ be the weight of $\boldsymbol{\pi}$, which is the product $\psi(\tau_1) \cdots \psi(\tau_j)$. Let $\boldsymbol{v}(\boldsymbol{\pi})$ be the top stack vector that would result at the end of $\boldsymbol{\pi}$. Let $\Pi[q_0, \bot, 0 \rightsquigarrow r, y, t]$ be the set of all runs that end at step $t$ in state $r$ and with $y$ on top of the stack. For each $r, y, t$, the dVPDA computes a summation over all of those runs.

$$\boldsymbol{r}_t[r, y] \stackrel{\text{def}}{=} \frac{\sum_{\boldsymbol{\pi} \in \Pi[q_0, \bot, 0 \rightsquigarrow r, y, t]} \psi(\boldsymbol{\pi}) \, \boldsymbol{v}(\boldsymbol{\pi})}{\sum_{r' \in Q} \sum_{y' \in \Gamma} \sum_{\boldsymbol{\pi} \in \Pi[q_0, \bot, 0 \rightsquigarrow r', y', t]} \psi(\boldsymbol{\pi})} \tag{6}$$

Again, this resembles a structured attention mechanism over syntactic structures. DuSell & Chiang (2023) show how to compute Eq. (6) efficiently using a dynamic programming algorithm. The stack reading $\boldsymbol{r}_t$ is the concatenation of all $\boldsymbol{r}_t[r, y]$. Note that Eq. (5) is a special case of Eq. (6) where $Q = \{q_0\}$, $\Gamma = \{\bot\}$, $\boldsymbol{v}_0 = \mathbf{0}$, transition weights are locally normalized to sum to 1, and replace transitions before the first push in a run have a weight of 1.

## B Details of neural network architectures

In this section, we describe each of the base neural network architectures (simple RNN, LSTM, and transformer) in more detail. All three are based on the built-in PyTorch implementations (Paszke et al., 2019) and closely follow the settings of Butoi et al. (2025). Each architecture consists of a configurable number of layers $L$. Each architecture uses a learned input embedding matrix $\boldsymbol{E}$ to map each input symbol $w_t$ of the input string to an embedding $\boldsymbol{x}_t = \boldsymbol{E}_{w_t}$. The size of the embeddings is $d$.

### B.1 Simple RNN

Let $\boldsymbol{h}_t^{(\ell)}$ denote the hidden state of the $\ell^{\text{th}}$ layer at timestep $t$. We apply dropout to the input embeddings, the hidden states between layers, and the hidden states of the last layer, following Zaremba et al. (2015). The initial hidden state of each layer is learned. The simple RNN architecture is defined as follows.

$$\boldsymbol{x}_t' \stackrel{\text{def}}{=} \text{DROPOUT}(\boldsymbol{x}_t) \qquad (1 \leq t \leq n) \tag{7a}$$

$$\boldsymbol{\hbar}_t^{(0)} \stackrel{\text{def}}{=} \boldsymbol{x}_t' \qquad (1 \leq t \leq n) \tag{7b}$$

$$\boldsymbol{h}_0^{(\ell)} \stackrel{\text{def}}{=} \tanh(\boldsymbol{w}_0^{(\ell)}) \qquad (1 \leq \ell \leq L) \tag{7c}$$

$$\boldsymbol{h}_t^{(\ell)} \stackrel{\text{def}}{=} \tanh\left(\boldsymbol{W}_{\text{h}}^{(\ell)} \begin{bmatrix} \boldsymbol{\hbar}_t^{(\ell-1)} \\ \boldsymbol{h}_{t-1}^{(\ell)} \end{bmatrix} + \boldsymbol{b}_{\text{h}}^{(\ell)}\right) \qquad (1 \leq \ell \leq L; 1 \leq t \leq n) \tag{7d}$$

$$\boldsymbol{\hbar}_t^{(\ell)} \stackrel{\text{def}}{=} \text{DROPOUT}(\boldsymbol{h}_t^{(\ell)}) \qquad (1 \leq \ell \leq L - 1; 1 \leq t \leq n) \tag{7e}$$

$$\mathcal{H}_t \stackrel{\text{def}}{=} (\boldsymbol{h}_t^{(1)}, \ldots, \boldsymbol{h}_t^{(L)}) \qquad (0 \leq t \leq n) \tag{7f}$$

$$\boldsymbol{y}_t' \stackrel{\text{def}}{=} \text{DROPOUT}(\boldsymbol{h}_t^{(L)}) \qquad (0 \leq t \leq n) \tag{7g}$$

The learned parameters are $\boldsymbol{w}_0^{(\ell)} \in \mathbb{R}^d$, $\boldsymbol{W}_{\mathrm{h}}^{(\ell)} \in \mathbb{R}^{d \times 2d}$, and $\boldsymbol{b}_{\mathrm{h}}^{(\ell)} \in \mathbb{R}^d$ ($1 \le \ell \le L$).

## B.2 LSTM

We apply dropout according to Zaremba et al. (2015). The initial hidden state of each layer is learned. The LSTM architecture is defined as follows. Let $\odot$ denote elementwise multiplication.

$$\boldsymbol{x}_t' \overset{\text{def}}{=} \text{DROPOUT}(\boldsymbol{x}_t) \qquad (1 \le t \le n) \tag{8a}$$

$$\boldsymbol{\hbar}_t^{(0)} \overset{\text{def}}{=} \boldsymbol{x}_t' \qquad (1 \le t \le n) \tag{8b}$$

$$\boldsymbol{h}_0^{(\ell)} \overset{\text{def}}{=} \tanh(\boldsymbol{w}_0^{(\ell)}) \qquad (1 \le \ell \le L) \tag{8c}$$

$$\boldsymbol{c}_0^{(\ell)} \overset{\text{def}}{=} \boldsymbol{0} \qquad (1 \le \ell \le L) \tag{8d}$$

$$\boldsymbol{i}_t^{(\ell)} \overset{\text{def}}{=} \sigma\left(\boldsymbol{W}_{\mathrm{i}}^{(\ell)} \begin{bmatrix} \boldsymbol{\hbar}_t^{(\ell-1)} \\ \boldsymbol{h}_{t-1}^{(\ell)} \end{bmatrix} + \boldsymbol{b}_{\mathrm{i}}^{(\ell)}\right) \qquad (1 \le \ell \le L; 1 \le t \le n) \tag{8e}$$

$$\boldsymbol{f}_t^{(\ell)} \overset{\text{def}}{=} \sigma\left(\boldsymbol{W}_{\mathrm{f}}^{(\ell)} \begin{bmatrix} \boldsymbol{\hbar}_t^{(\ell-1)} \\ \boldsymbol{h}_{t-1}^{(\ell)} \end{bmatrix} + \boldsymbol{b}_{\mathrm{f}}^{(\ell)}\right) \qquad (1 \le \ell \le L; 1 \le t \le n) \tag{8f}$$

$$\boldsymbol{g}_t^{(\ell)} \overset{\text{def}}{=} \tanh\left(\boldsymbol{W}_{\mathrm{g}}^{(\ell)} \begin{bmatrix} \boldsymbol{\hbar}_t^{(\ell-1)} \\ \boldsymbol{h}_{t-1}^{(\ell)} \end{bmatrix} + \boldsymbol{b}_{\mathrm{g}}^{(\ell)}\right) \qquad (1 \le \ell \le L; 1 \le t \le n) \tag{8g}$$

$$\boldsymbol{o}_t^{(\ell)} \overset{\text{def}}{=} \sigma\left(\boldsymbol{W}_{\mathrm{o}}^{(\ell)} \begin{bmatrix} \boldsymbol{\hbar}_t^{(\ell-1)} \\ \boldsymbol{h}_{t-1}^{(\ell)} \end{bmatrix} + \boldsymbol{b}_{\mathrm{o}}^{(\ell)}\right) \qquad (1 \le \ell \le L; 1 \le t \le n) \tag{8h}$$

$$\boldsymbol{c}_t^{(\ell)} \overset{\text{def}}{=} \boldsymbol{f}_t^{(\ell)} \odot \boldsymbol{c}_{t-1}^{(\ell)} + \boldsymbol{i}_t^{(\ell)} \odot \boldsymbol{g}_t^{(\ell)} \qquad (1 \le \ell \le L; 1 \le t \le n) \tag{8i}$$

$$\boldsymbol{h}_t^{(\ell)} \overset{\text{def}}{=} \boldsymbol{o}_t^{(\ell)} \odot \tanh(\boldsymbol{c}_t^{(\ell)}) \qquad (1 \le \ell \le L; 1 \le t \le n) \tag{8j}$$

$$\boldsymbol{\hbar}_t^{(\ell)} \overset{\text{def}}{=} \text{DROPOUT}(\boldsymbol{h}_t^{(\ell)}) \qquad (1 \le \ell - 1 \le L; 1 \le t \le n) \tag{8k}$$

$$\mathcal{H}_t \overset{\text{def}}{=} (\boldsymbol{h}_t^{(1)}, \boldsymbol{c}_t^{(1)} \ldots, \boldsymbol{h}_t^{(L)}, \boldsymbol{c}_t^{(L)}) \qquad (0 \le t \le n) \tag{8l}$$

$$\boldsymbol{y}_t' \overset{\text{def}}{=} \text{DROPOUT}(\boldsymbol{h}_t^{(L)}) \qquad (0 \le t \le n) \tag{8m}$$

The learned parameters are $\boldsymbol{w}_0^{(\ell)} \in \mathbb{R}^d$, $\boldsymbol{W}_{\mathrm{i}}^{(\ell)} \in \mathbb{R}^{d \times 2d}$, $\boldsymbol{b}_{\mathrm{i}}^{(\ell)} \in \mathbb{R}^d$, $\boldsymbol{W}_{\mathrm{f}}^{(\ell)} \in \mathbb{R}^{d \times 2d}$, $\boldsymbol{b}_{\mathrm{f}}^{(\ell)} \in \mathbb{R}^d$, $\boldsymbol{W}_{\mathrm{g}}^{(\ell)} \in \mathbb{R}^{d \times 2d}$, $\boldsymbol{b}_{\mathrm{g}}^{(\ell)} \in \mathbb{R}^d$, $\boldsymbol{W}_{\mathrm{o}}^{(\ell)} \in \mathbb{R}^{d \times 2d}$, and $\boldsymbol{b}_{\mathrm{o}}^{(\ell)} \in \mathbb{R}^d$ ($1 \le \ell \le L$).

## B.3 TRANSFORMER

Our transformer implementation is based on that of PyTorch. Following Vaswani et al. (2017), we map input symbols to vectors of size $d$ with a scaled embedding layer and add sinusoidal positional encodings. We always prepend a reserved BOS token whose corresponding output is used as the probabilities for the first token. We tie the input and output embeddings (Press & Wolf, 2017). We set the number of hidden units in each feedforward layer to $2d$. We use pre-norm instead of post-norm and apply layer norm to the output of the last layer. We use the same dropout rate throughout the transformer. We apply it in the same places as Vaswani et al. (2017), and, as implemented by PyTorch, we also apply it to the hidden units of feedforward sublayers and to the attention probabilities of scaled dot-product attention operations.

## C EXPERIMENTAL DETAILS

Here, we provide more details about the experiments in §5.

For all models, wherever dropout is applicable, we use a dropout rate of 0.1. In our transformers, all of the scaled dot-product attention sublayers are causally masked and use 4 heads.

Following DuSell & Chiang (2024), we initialize all fully-connected layers with Xavier uniform initialization (Glorot & Bengio, 2010), except for layers involved in the recurrent update of RNNs

and LSTMs and in the standard scaled dot-product attention layers in transformers. For layer norm, we initialize all weights to 1 and all biases to 0. We initialize all other parameters by sampling uniformly from $[-0.1, 0.1]$.

We group examples of similar lengths into minibatches. We limit the total number of tokens per minibatch, including BOS, EOS, and padding tokens, by a hyperparameter $B$. For each string $w$ in each minibatch, we compute $-\log p_M(w)$ and take the mean over all $w$ in the batch to form the loss function that we minimize. We optimize all models with Adam.[3] We use $L^2$ gradient clipping with a threshold of 10. We use cross-entropy of whole strings $w$ in the validation set, micro-averaged by the length of $w$ plus EOS, to control the learning rate schedule and early stopping. We measure this at regular checkpoints, which we take every 80k training examples. We multiply the learning rate by 0.5 whenever the validation cross-entropy does not improve after 2 checkpoints. We stop early when the validation cross-entropy does not improve after 3 checkpoints. When evaluating a model, we always use the parameters corresponding to the checkpoint that resulted in the lowest validation cross-entropy during training.

During our hyperparamater search, we randomly sample $B$ from a uniform distribution over $[512, 2048]$, and we randomly sample the initial learning rate from a log-uniform distribution over $[10^{-5}, 10^{-3}]$.

---

[3]We found that using AdamW (Adam with weight decay) resulted in generally *weaker*, not stronger, hierarchical generalization.

