# OpenReview forum: "Bearing Syntactic Fruit with Stack-Augmented Neural Networks"
_ICLR.cc/2026/Conference — Submitted to ICLR 2026_

### Official Review · Reviewer_4ng3 · 2025-10-29

**Soundness:** 2
**Presentation:** 3
**Contribution:** 2
**Rating:** 4
**Confidence:** 3

**Summary:**

The paper investigates whether neural networks with explicit stack-based memory exhibit human-like hierarchical generalization under poverty-of-the-stimulus conditions. It evaluates stack-augmented simple RNNs, LSTMs, and Transformers on two controlled tasks (English question formation and tense reinflection). The models implement two differentiable stack mechanisms: the superposition stack (a continuous relaxation over push / no-op / pop) and a nondeterministic differentiable vector pushdown automaton (dVPDA), trained end-to-end. On question formation, Transformers with nondeterministic stack attention show stronger hierarchical generalization (avg over 5 runs, Table 1), with ≈32% conditional probability of the full correct output (Tf+Nd) and ≈86% fine-grained accuracy (Tf+Nd+Nd) on the generalization set, compared to the vanilla Transformer (≈0.5% CP and ≈0.65 FA). In contrast, similar hierarchical generalization does not reliably emerge on tense reinflection. The authors release code and a Docker environment to reproduce the experiments.

**Strengths:**

The paper tackles an issue at the intersection of machine learning and linguistic theory — whether explicit structural memory can induce human-like syntactic generalization under “poverty of the stimulus.” The motivation is articulated clearly and grounded in prior psycholinguistic work. The work effectively connects formal-language theory, ML architectures, and cognitive modeling. This synthesis gives the study conceptual depth beyond architecture comparisons.
The narrative is well structured: the motivation and results sections are especially clear. The presentation demonstrates awareness of both ML and linguistic fields. The proposed architecture is accompanied by substantial mathematical derivations that ground the approach formally.
Datasets are precisely controlled via PCFG generation, and the evaluation setup directly parallels established human-language acquisition experiments. The authors provide full reproducibility artifacts, which substantially increases the paper’s value for the community and sets a reproducibility standard for cognitively oriented modeling work.

**Weaknesses:**

While the paper presents an elegant and well-executed study linking stack-augmented architectures with hierarchical generalization, several limitations remain.

First, the experimental scope is narrowed: all results rely on small synthetic grammars and two tasks within the poverty-of-stimulus framework, leaving unclear whether the observed effects generalize to naturalistic or cross-linguistic data. The paper lacks discussion of other diagnostic cases traditionally used to probe structure dependence — such as reflexive binding, agreement attraction, or relative clause attachment. Outlining or testing additional paradigms would strengthen the claim that stack-augmented architectures capture a generalizable linguistic bias rather than a task-specific one.

Second, the mathematical exposition of the differentiable stacks is overly dense for a broad audience; the intuition behind key derivations is under-explained (a high-level diagram is present (Figure 4), but it does not unpack the derivations), limiting accessibility.

Third, ablation analysis is insufficiently targeted: without probing stack hyperparameters (depth/size, degree of nondeterminism) and isolating the effect of shortcut connections, it is difficult to determine which architectural factors drive the gains. Though std on 5 runs are provided along with the final scores, the absence of statistical significance testing makes it hard to assess robustness of the reported improvements.

Finally, the failure on the tense-reinflection task and the limited ethical discussion of English-centric bias somewhat weaken the paper’s broader psycholinguistic claims. The failure to generalize on tense reinflection is under-discussed; understanding this discrepancy would critically test the authors’ hypothesis. Overall, the link from architectural bias to human acquisition mechanisms feels suggestive but somewhat overstated given the current evidence base.

The paper’s psycholinguistic framing would benefit from connecting to existing syntactic evaluation frameworks such as BLiMP, which operationalize a broad set of “poverty-of-stimulus” diagnostics through minimal-pair tests.

**Questions:**

1) It is noted that only Transformer variants successfully learn the in-distribution test set for both tasks. Could the authors elaborate on why recurrent and stack-augmented recurrent models fail to do so — e.g., optimization difficulties, hyperparameter sensitivity, or fundamental capacity limitations? A short discussion would clarify how comparable the models truly are.

2) Means and standard deviations over five runs are reported, but no significance testing is presented. Could the authors provide or comment on statistical significance of the main improvements (e.g., CP, FA, Log Ratio), to support that observed gains are not due to random variation?

3) The study focuses on two syntactic transformations within the poverty-of-the-stimulus framework. Are there plans to test additional syntactic phenomena or more naturalistic datasets to assess how general the observed bias is?

4) While the authors compare several model variants, it remains unclear which architectural or training factors (e.g., nondeterminism, stack depth, shortcut connections) primarily drive the hierarchical generalization. Could the authors briefly describe or plan targeted ablations to isolate these effects?

5) The model’s failure on the tense-reinflection task is intriguing. Could you expand on possible reasons—task properties, model bias, or data setup—and whether alternative training regimes change this outcome?

---

> ### Author Response · Authors · 2025-11-18
>
> Dear Reviewer 4ng3,
>
> Thank you very much for your detailed review and your suggestions for improving our work! We look forward to engaging with you to address your concerns.
>
> > Second, the mathematical exposition of the differentiable stacks is overly dense...
>
> Thank you for pointing this out. We'll work on condensing this material.
>
> > Finally, the failure on the tense-reinflection task and the limited ethical discussion of English-centric bias somewhat weaken the paper’s broader psycholinguistic claims...
>
> A handful of other poverty-of-the-stimulus style datasets beyond question formation and tense reinflection have been used in prior work, including other languages such as German. If we report back with results on these datasets, would this alleviate your concerns?
>
> We will also work on adding more discussion of the failure to generalize on tense reinflection.
>
> > The paper’s psycholinguistic framing would benefit from connecting to existing syntactic evaluation frameworks such as BLiMP...
>
> Thanks for this suggestion; we would be happy to better contextualize our work in relation to BLiMP. Our tasks focus specifically on syntactic generalization, whereas BLiMP targets multiple phenomena, namely syntax, morphology, and semantics. However, BLiMP doesn't control for the poverty of the stimulus in the same way that our tasks do, since it does not place restrictions on whether disambiguating examples can appear in the training data. Our tasks and BLiMP essentially answer different scientific questions: our tasks test whether a learning algorithm has a hierarchical inductive bias, whereas BLiMP tests whether a pre-trained LM has mastered certain long-tail phenomena, which it may very well have encountered in its training data.
>
> (continued in next comment...)

---

> > ### Author Response · Authors · 2025-11-18
> >
> > ## Answers to Questions
> >
> > 1. We believe there's a simple explanation: the information bottleneck caused by the constant size of the hidden state in the RNN and LSTM. In order for our RNN and LSTM language models to transform the input into the output, they need to memorize most of the input string using the hidden state, and this tends to be lossy. The transformer language model, on the other hand, does not have this bottleneck because of its attention mechanism. We'll add this discussion to the paper.
> > 2. We'll run a paired permutation test using Monte Carlo sampling and the tmax method to account for multiple variables, based on [this implementation](https://mne.tools/stable/generated/mne.stats.permutation_t_test.html). Would this satisfy your concern?
> > 3. We agree that broadening our experiments would strengthen the claims of the paper. As mentioned above, we can report back with results on other poverty-of-the-stimulus tasks used in prior work.
> >
> >     Naturalistic datasets are a bit more challenging to work with if our goal is to test the poverty of the stimulus, since we need to control whether disambiguating examples for a specific phenomenon appear in the training data. One idea is to train an n-gram model and a context-free grammar on an annotated treebank, such as the Penn Treebank, construct a training set that is consistent with both the n-gram model and CFG, and evaluate on data that is consistently only with the CFG. Would that alleviate your concerns?
> >
> >     Another idea is a synthetic experiment based on regular languages vs. non-regular context-free languages, which cuts to the heart of linear vs. hierarchical generalization. We could generate a dataset that is consistent with both a regular language and a non-regular but context-free language and see whether a model trained on that dataset generalizes according to the non-regular language. Would this also be of interest to you?
> > 4. We can certainly explore this more. The three stack hyperparameters are the vector size $m$ (Sup and Nd), the number of states $|Q|$ (Nd only), and stack alphabet size $|\Gamma|$ (Nd only). For computational cost reasons, $m$ needs to be small for Nd. We can run targeted experiments that vary these hyperparameters while keeping all other hyperparameters (including the total number of parameters) the same. We propose sweeping over
> >
> >     * $m = 25, 50, 75$ for Sup
> >     * $m = 2, 3, 4$ for Nd
> >     * $|Q| = |\Gamma| = 2, 3, 4$ for Nd
> >
> >     Given the large number of experiments, we might need to limit this to question formation. Would this satisfy your concerns?
> >
> >     As for the +R modification, Tables 1 and 2 essentially include ablations already, since each +R architecture has a corresponding version without +R. Did you have something more in mind?
> > 5. One reason may be that, unlike the question formation task, the tense reinflection task does not use auxiliary *do* for its verbs. This means the model needs to learn a separate pair of inflections for each verb, which may make it harder to disentangle lexical and syntactic information during training. McCoy et al. (2020) provide a variant of this dataset that uses auxiliary *do*, and we can test this hypothesis by re-running the experiments on this dataset.
> >
> >     Ahuja et al. (2025) showed that the language modeling training regime, which we use in this paper, is the best among several possible training regimes at promoting hierarchical generalization. This makes quite a bit of sense, because applying the language modeling objective to the input encourages the model to share representations of the input syntax between, e.g., the DECL and QUEST examples. DECL examples include distractors, whereas QUEST examples do not, so this may help the model anticipate distractors in the QUEST examples. So, it seems unlikely that different training regimes would improve hierarchical generalization.
> >
> >     One thing we did not yet explore in this paper is sequence-to-sequence architectures augmented with stacks. Although Ahuja et al. (2025) showed that this training regime underperforms language modeling for standard architectures, this might not hold for the stack-augmented architectures. Would you be interested in seeing additional experiments on sequence-to-sequence models?

---

### Official Review · Reviewer_6mf3 · 2025-11-01

**Soundness:** 3
**Presentation:** 3
**Contribution:** 1
**Rating:** 2
**Confidence:** 3

**Summary:**

This work intends to investigate whether neural network models of various architectures can be augmented with a stack mechanism in order to imbue the model with an inductive bias towards hierarchical syntactic generalization. The authors investigate a number of architectural permutations and two different stack mechanisms on a question formation and tense reinflection task, finding that a transformer with a stack mechanism imbues a hierarchical syntactic inductive bias in the question formation task (but not the tense task).

**Strengths:**

This work rigorously sweeps over architectural parameterizations.

The presentation of the existing methods is thorough.

**Weaknesses:**

The primary weakness of this work is that the empirical contribution is relatively small. The majority of the text is dedicated to explaining architectural details already described in related work, or in describing fairly small architectural innovations that do not yield empirical improvements (i.e., the +R reading shortcut still results in a negative LR). Many of these descriptions can be put in an appendix, with the main text expanded to include additional analyses and experiments.

For example, it would be interesting to know whether there were any systematic differences between questions that resulted in hierarchical generalization and those that did not. It would also be exciting to explore combinations of stack networks with existing findings, like Ahuja et al.’s overtrained transformer result. Perhaps one of the stack transformers gets to the overtrained regime faster than the standard transformer?

To be clear, the underlying question and research direction are exciting, but the work needs further empirical development prior to publication.

**Questions:**

Typos:

348 Ues
177 inptu

---

> ### Author Response · Authors · 2025-11-18
>
> Dear Reviewer 6mf3,
>
> Thank you very much for your review and your suggestions for improving the contributions of the paper. We'll be happy to work with you to expand our empirical contributions.
>
> > The majority of the text is dedicated to explaining architectural details already described in related work...
>
> This is a fair criticism, and we'll work on condensing the background material to make room for more experiments and analysis.
>
> > For example, it would be interesting to know whether there were any systematic differences between questions that resulted in hierarchical generalization and those that did not.
>
> That's a good idea. We'll take a look at this and get back to you.
>
> > Perhaps one of the stack transformers gets to the overtrained regime faster than the standard transformer?
>
> We'll take a look at this too and get back to you.
>
> In the meantime, we're happy to brainstorm other experiments and analyses that would bolster our contributions. Some of the suggestions from the other reviewers were:
>
> 1. Inspecting the learned stack actions to see what the models learned about syntactic structure.
> 2. Running experiments on additional poverty-of-the-stimulus tasks proposed in prior work.
> 3. Exploring the effect of stack size hyperparameters on hierarchical generalization.
> 4. Running experiments on more naturalistic data, perhaps a task generated using n-gram models vs. context-free grammars extracted from the Penn Treebank.
>
> Would any of these suggestions increase your confidence in our results?

---

### Official Review · Reviewer_TGjH · 2025-11-03

**Soundness:** 3
**Presentation:** 2
**Contribution:** 3
**Rating:** 6
**Confidence:** 4

**Summary:**

This paper explores whether stack-augmented Transformer and RNN architectures (specifically superposition stack of Joulin & Mikolov 2015 and (modifications of) the  nondeterministic stack of DuSell & Chiang 2024) leads to improved hierarchical generalization on question formation and tense reinflection tasks that are designed to tease apart the learner's biases towards linear vs. hierarchical generalization. The explored modifications to DuSell & Chiang's architecture is minor: for recurrent models, there's a direct feed-in ("short circuit") added to the output of the network at each timestep from the timestep's stack to avoid an off-by-one lag, and adding two stack attention layers to transformers. The general finding is that networks augmented with stacks show increase in their preference for hierarchical generalization for question formation for Transformers and LSTMs (but not RNNs), but not for tense reinflection. Furthermore, for LSTMs to benefit from stacks, the short circuiting mechanism seems necessary.

The paper's contribution is straightforward - it directly continues the exploration of the research question "Which factors make neural networks prefer hierarchical generalization (as operationalized by the question formation and tense reinflection datasets)?" by various work cited in the paragraph starting from L071. The findings are unsurprising (not in a negative way at all, just in the sense that it would be have been more surprising had the trends been different) but sufficiently interesting. I have a few clarification questions regarding the comparisons of results to existing work and how related work's contributions are represented, which I believe could be fruitfully resolved during the discussion.

**Strengths:**

- The research question, the hypothesis, and the takeaways are clear and straightforward to understand, partly owing to the fact that this problem operationalized in terms of the specific datasets used has a well established cottage industry (again not in a negative sense at all, just for lack of a better term)
- The exposition of the scope of the contribution and the experimental setup is generally clear.

**Weaknesses:**

- I think "syntactically supervised" is a bit of a vague term for addressing what McCoy et al. 2020 did - my inference is that this refers to the tree-structured network experiments, but highlighting the fact that training of such networks require explicit representations of syntactic structure in the training data. Since "syntactically supervised" is ambiguous between only data-level supervision and the architecture requiring parsed inputs, it might be expositionally clearer & make the new contribution that this paper clearer to say out loud that tree-structured networks have been explored as a way of endowing structural inductive biases, but this also requires the input to be fully parsed. It would also be informative to compare against this result in Table 1 & 2.
- I think it is not entirely accurate to lump Yedetore & Kim (2024) under "trained long past convergence on the validation data" although they do report grokking results, since the main claim of that paper is about an auxiliary objective of form-to-meaning mapping task, which leads to preference for hierarchical generalization without any of the listed changes. I think this result actually should put a qualifier on the results presented in this paper: for instance, the claim in the abstract "that they do so only under special conditions: when syntactically supervised, when pre-trained on massive corpora, or when trained long past convergence. In this paper, we demonstrate, for the first time, neural network architectures that are able to generalize in human-like fashion without any of the aforementioned requirements: stack-augmented neural networks." should be qualified such that "for the first time" claim really is "for the first time in networks that are purely trained on surface forms". Their results are also a lot stronger than the best FA reported in the results section ("Furthermore, Tf+Nd and Tf+Nd+Nd attain the highest FA (up to 86%) [...] It also surpasses the approximately 76% FA of Ahuja et al.’s (2025) overtrained transformer language model.") - closer to 100% FA.
- Continuing the comparison to prior work discussion, on L403: "For both tasks, only transformers learn the in-distribution test set" If I remember correctly, the McCoy et al. 2020 paper reported that LSTMs do get near ceiling full-sentence accuracy on the in-distribution test set. Does this suggest that the LSTM training here is somehow degenerate, or is there an alternative explanation for this?
- L405: "This also improves over the 10% of Mulligan et al.’s (2021) GRU trained with multi-task learning." I am also somewhat confused by this claim because their Figure 1 reports close to 50% full sentence accuracy with a multitask trained GRU. Am I misreading something here?

**Questions:**

- I am curious whether you did any analyses beyond quantitative results about generalization preference, and wonder in general the newly introduced architectural components enable any interesting analyses (even if you haven't looked into this) like seeing if the right structures were indeed inferred.

I think I put most other contentful questions in the Weaknesses section, so these are just small comments/suggstions:

- L96- "The question, then, is really whether a reasonably simple learning algorithm—not the kind of contrived example just mentioned, but perhaps a neural network architecture with minimal assumptions about the specific task—can learn a rule like MOVE-MAIN from ambiguous data while still attaining competitive performance on natural language benchmarks." I disagree, if what we're interested in is cognitive modeling, competitive performance on natural language benchmarks are irrelevant. But maybe by "natural language benchmarks" what is intended is only a subset of them that is relevant to cognitive modeling.
- L98- "We offer stack-augmented neural networks as a positive example that hierarchical generalization need not originate from the training data alone." While I get the intent of the statement, "need not" is a bit of a strange way to phrase things, since this is exactly aligned with the poverty of the stimulus claim, i.e., hierarchical generalization needs to originate from training data-external means.
- L348: ues -> use
- "For transformers with one stack attention layer, we swap it into the attention mechanism of the third layer. For transformers with two stack attention layers, we swap them into the second and fourth layers." --- is the choice of swapped-in layers basically a hyperparameter?
- Presentation suggestion: Tables 1 and 2 are a bit overwhelming, maybe better to have the Tables in the Appendix and have a graphical visualization.

---

> ### Author Response · Authors · 2025-11-18
>
> Dear Reviewer TGjH,
>
> Thank you very much for your thorough and thoughtful review! We appreciate your familiarity with prior work and agree with many of the points you raised. We look forward to engaging with you to address them.
>
> > I think "syntactically supervised" is a bit of a vague term for addressing what McCoy et al. 2020 did...
>
> Yes, we were referring to the tree LSTM experiments of McCoy et al. (2020). We used "syntactically supervised" in the broad sense of "requires explicit parse trees in the data," but you are right to point out that what they did goes deeper than, say, including brackets as part of the input string, and actually determines the shape of the network's computation graph. We would be happy to make this clearer in the text and include a comparison in the main tables.
>
> > I think it is not entirely accurate to lump Yedetore & Kim (2024) under "trained long past convergence on the validation data"...
>
> We'll update the text to avoid making that implication. We cited that paper as an example of grokking simply because, as you pointed out, they included grokking results, albeit that was not the focus of their paper. We agree that the phrasing "for the first time in networks that are purely trained on surface forms" does a good job of characterizing what we did, and we'll update that too.
>
> > Their results are also a lot stronger than the best FA reported in the results section ("Furthermore, Tf+Nd and Tf+Nd+Nd attain the highest FA (up to 86%) [...] It also surpasses the approximately 76% FA of Ahuja et al.’s (2025) overtrained transformer language model.") - closer to 100% FA.
>
> Can you clarify which results in Ahuja et al. (2025) you're referring to? We are referring to their results in Section 3.3 with the transformer LM on the generalization set for question formation, where they report a mean FA of roughly 76% (although they mention that some seeds get 100%). These results are also shown in their Figure 1, row: Generalization Accuracy, column: Question Formation, red bar.
>
> > ...Does this suggest that the LSTM training here is somehow degenerate, or is there an alternative explanation for this?
>
> We believe there's a simple explanation: McCoy et al. (2020) used LSTM encoder-decoder networks with attention, whereas we use LSTM language models without attention. In our LSTM language model, there's an information bottleneck due to its fixed-size hidden state, which makes it harder for the LSTM to memorize parts of the input and copy them to the output faithfully. The encoder-decoder with attention doesn't have this bottleneck.
>
> > L405: "This also improves over the 10% of Mulligan et al.’s (2021) GRU trained with multi-task learning." I am also somewhat confused by this claim...
>
> This is a mistake on our part -- thank you very much for catching this. We were referring to their GRU result *without* multi-task learning, shown in their Figure 1, column: Full Sentence, row: Question Formation, gray bar, which gets about 10%. We mistakenly identified this as the GRU trained on multiple tasks. We will fix this.
>
> > I am curious whether you did any analyses beyond quantitative results about generalization preference...
>
> We have not, but we are happy to do so during the rebuttal period. In principle, it is possible to inspect the learned stack actions to get a sense of the learned syntactic structure. This is fairly straightforward for the +Sup stack, which only has 3 actions, but it is more complicated for the +Nd stack, which has a large number of actions. We will take a look at this and get back to you.
>
> Reviewer 6mf3 suggested looking for systematic differences between questions that result in hierarchical generalization and those that do not. Would this also be of interest to you?
>
> > L96 ... I disagree, if what we're interested in is cognitive modeling, competitive performance on natural language benchmarks are irrelevant...
>
> We probably agree on this, in the sense that many natural language benchmarks can effectively be gamed by heuristics that have little to do with cognitive modeling. We're interested in long-tail phenomena that would benefit from better models of human language processing. As people who are interested in using cognitive modeling to improve NLP systems, though, we also have an interest in models that maintain competitive performance on NLP tasks and also happen to be better cognitive models.
>
> > L98 ..."need not" is a bit of a strange way to phrase things, since this is exactly aligned with the poverty of the stimulus claim...
>
> Would this be better? "We offer stack-augmented neural networks as a positive example that hierarchical generalization can emerge without explicit cues in the training data."
>
> > ...is the choice of swapped-in layers basically a hyperparameter?
>
> Yes, they can be swapped in anywhere.
>
> > Presentation suggestion: Tables 1 and 2 are a bit overwhelming...
>
> Thank you for this suggestion. We'll think of ways of presenting this in a more digestible way.

---

### Author Response · Authors · 2025-12-03
**Summary of Changes**

Dear AC,

For your convenience, here is a summary of some of the concerns that the reviewers raised and what we have done to address them. For simplicity, we will refer to reviewers TGjH, 6mf3, and 4ng3 as R1, R2, and R3, respectively.

* R1 had particular comments about the writing, namely about the characterization of McCoy et al. (2020)'s Tree-GRU experiments as being syntactically supervised, Yedetore & Kim (2024) being lumped in with the work on grokking, qualifying our results as being the best of models trained purely on surface forms, and the phrasing of "need not" at the end of the introduction. We have fixed all of these issues in the updated draft.
* R1 suggested adding the Tree-GRU results of McCoy et al. (2020) to our tables of results, so we have added them.
* R1 found a mistake in how we reported one of the results from Mulligan et al. (2021). We have removed this.
* R1, R2, and R3 all suggested condensing the background material on differentiable stacks and moving some of it to the appendix. We have now condensed it considerably.
* R3 requested that we discuss the relevance of our tasks to BLiMP. As we pointed out in our response, BLiMP tests a very different scientific question, so it is not directly relevant to our work.
* R1 and R3 wanted to know why the RNN and LSTM models do not learn the in-distribution test set. We have added an explanation to the paper of why this happened.
* R2 suggested analyzing our results through the lens of the "grokking" finding of Murty et al. (2023). Murty et al. (2023) showed that even after the validation performance of the transformer converges during training, its generalization performance continues to improve for a long time afterwards, i.e., it starts "grokking" the hierarchical structure of the task. R2 speculated that the Tf+Nd model might reach this grokking regime earlier. This was already somewhat implied by the fact that Tf+Nd gets higher accuracy than Tf without needing to be overtrained, but we ran experiments to further vet this hypothesis.

  Surprisingly, we have been unable to reproduce the grokking results of Murty et al. (2023) for even the baseline transformer, despite matching their hyperparameter settings very closely (for example, we use the same learning rate scheduler and the optimal number of transformer layers they reported). Instead, across several random seeds, we have seen that the fine accuracy of the model has already plateaued by the time it would have stopped early. A slight difference is that we use validation *cross-entropy* instead of accuracy as the early stopping criterion. We also use different parameter initialization. We have seen the same results when running experiments using the same `ReduceLROnPlateau` scheduler used in our main experiments. Note that although Yedetore & Kim (2024) and Ahuja et al. (2025) reproduced the grokking finding, their experiments used the same codebase as Murty et al. (2023), whereas we use an entirely different codebase. To our knowledge, no one has tried reimplementing this finding from scratch before. This suggests that the grokking finding is not very robust to begin with, but sensitive to low-level details in the experimental setup. The difficulty of reproducing the grokking phenomenon with baseline transformers further highlights the significance or our results with stack-augmented transformers, which do not even require overtraining to generalize hierarchically.
* R3 inquired as to why the models did not generalize hierarchically on the tense reinflection. One explanation may be that the task does not use auxiliary *do*, so we have started experiments on a version of this task that does use auxiliary *do*. These experiments are still in progress, but we will be able to include them in a future draft.

---

### Meta-Review · Area_Chair_fo6j · 2026-01-14

**Summary:**

The paper tackles an issue at the intersection of machine learning and linguistic theory — whether explicit structural memory can induce human-like syntactic generalization under “poverty of the stimulus.” The motivation is articulated clearly and grounded in prior psycholinguistic work. The work effectively connects formal-language theory, ML architectures, and cognitive modeling. This synthesis gives the study conceptual depth beyond architecture comparisons. The narrative is well structured: the motivation and results sections are especially clear. The presentation demonstrates awareness of both ML and linguistic fields. The proposed architecture is accompanied by substantial mathematical derivations that ground the approach formally. Datasets are precisely controlled via PCFG generation, and the evaluation setup directly parallels established human-language acquisition experiments. The authors provide full reproducibility artifacts, which substantially increases the paper’s value for the community and sets a reproducibility standard for cognitively oriented modeling work.

Overall, the concerns included:
* The primary weakness of this work is that the empirical contribution is relatively small.
* The experimental scope is narrowed: all results rely on small synthetic grammars and two tasks within the poverty-of-stimulus framework, leaving unclear whether the observed effects generalize to naturalistic or cross-linguistic data.
* The mathematical exposition of the differentiable stacks is overly dense for a broad audience; the intuition behind key derivations is under-explained (a high-level diagram is present (Figure 4), but it does not unpack the derivations), limiting accessibility.
* Ablation analysis is insufficiently targeted: without probing stack hyperparameters (depth/size, degree of nondeterminism) and isolating the effect of shortcut connections, it is difficult to determine which architectural factors drive the gains.
* The failure on the tense-reinflection task and the limited ethical discussion of English-centric bias somewhat weaken the paper’s broader psycholinguistic claims.

**Reviewer Concerns:**

The authors promised to address many of the concerns, but were light on details in most cases and I think many of them are still outstanding in the sense that they would require significant new experiments to address.

**Reviewer Scores:**

Reviewer TGjH has some confusions that were clarified, but the other more negative reviewers had substantial concerns that were left open.

---

### Decision · Program_Chairs · 2026-01-26

Reject